

# Modelling micro- and macrophysical contributors to the dissipation of an Arctic mixed-phase cloud during the Arctic Summer Cloud Ocean Study (ASCOS)

Katharina Loewe[1], Annica M. L. Ekman[2], Marco Paukert[1], Joseph Sedlar[3], Michael Tjernström[2], and Corinna Hoose[1]

[1]Institute of Meteorology and Climate Research, Karlsruhe Institute of Technology, Karslruhe, Germany
[2]Department of Meteorology and Bolin Centre for Climate Research, Stockholm University, Stockholm, Sweden
[3]Swedish Meteorological Hydrological Institute, Norrköping, Sweden

*Correspondence to:* Katharina Loewe (katharina.loewe@kit.edu)

**Abstract.** The Arctic climate is changing; temperature changes in the Arctic are greater than at mid-latitudes, and changing atmospheric conditions influence Arctic mixed-phase clouds, which are important for the Arctic surface energy budget. These low-level clouds are frequently observed across the Arctic. They impact the turbulent and radiative heating of the open water, snow and sea-ice covered surfaces, and are influencing the boundary layer structure. Therefore the processes that affect mixed-

phase cloud life cycles are extremely important, yet relatively poorly understood. In this study, we present sensitivity studies using semi-idealized large eddy simulations (LES) to identify processes contributing to the dissipation of Arctic mixed-phase clouds. We found that one potential main contributor to the dissipation of an observed Arctic mixed-phase cloud, during the Arctic Summer Cloud Ocean Study (ASCOS) field campaign, was a low cloud droplet number concentration (CDNC) of about $2\,\mathrm{cm^{-3}}$. Introducing a high ice crystal concentration of $10\,\mathrm{l^{-1}}$ also resulted in cloud dissipation, but such high ice crystal

concentrations were deemed unlikely for the present case. Sensitivity studies simulating the advection of dry-air above the boundary layer inversion, as well as a modest increase in ice crystal concentration of $1\,\mathrm{l^{-1}}$, did not lead to cloud dissipation. As a requirement for small droplet numbers, pristine aerosol conditions in the Arctic environment are therefore considered an important factor determining the lifetime of Arctic mixed-phase clouds.

## 1 Introduction

The Arctic is a unique region that is highly sensitive to changes in climate (Curry et al., 1996). Since the mid 1960's, the Arctic annual average temperature has increased about at least twice as fast as the global average (Serreze and Barry, 2011), and the sea ice has seen a rapid decrease in all seasons, especially in summer; the annual summer minimum sea ice extent has been dropping by around $12\,\%$ per decade (Serreze and Stroeve, 2015; Vaughan et al., 2013).

Clouds play an essential role for the surface energy budget due to their ability to absorb and reflect radiation. Depending on the

physical properties of the clouds, they radiatively cool or warm the atmosphere and surface. Mixed-phase clouds, containing supercooled liquid and ice crystals simultaneously, are common in the Arctic. They occur more than $40\,\%$ of the time and





are most frequent in the spring and autumn transition seasons (Shupe et al., 2006, 2011). Arctic mixed-phase clouds differ from lower latitude mixed-phase clouds in that a) they may persist for several days (Morrison et al., 2012); and b) low-level temperature inversions, common to the Arctic boundary layer, do not always mark the vertical extent of the cloud layer (Sedlar et al., 2012). Additionally, ice crystals tend to form within the relatively thin layer of supercooled water and precipitate through

the sub-cloud layer (Morrison et al., 2012).

Concentrations of cloud condensation nuclei (CCN) are generally low in the Arctic, reaching values as low as $1\,cm^{-3}$ (Bigg and Leck, 2001; Mauritsen et al., 2011; Tjernström et al., 2014). Generally, the cloud droplet number concentration (CDNC) is lower than $100\,cm^{-3}$ (Hobbs and Rangno, 1998). Under such low aerosol and resulting CCN concentrations, a small increase in aerosol concentration can therefore drastically impact Arctic cloudiness and cloud radiative properties, increasing the surface

cloud radiative effect (Mauritsen et al., 2011). Ice nuclei (IN) concentrations are also relatively low in the Arctic (Pinto, 1998). Even a small increase in ice concentration can result in a transformation to an ice-only cloud, causing a dramatic change in the surface energy budget (Prenni et al., 2007). The concentration of measured IN can be uncertain due to different instrumentation and measurement techniques or due to natural variability of IN. Concentrations of IN can range as low as 0.01 to $1\,l^{-1}$, but concentrations can also reach even two to three orders of magnitude higher (Morrison et al., 2005; Rogers et al., 2001). Rogers

et al. (2001) found that persistent low-level stratus clouds contain low ice crystal concentrations, which indicates a low IN concentration in these clouds. The fact that an Arctic mixed-phase cloud with low CDNC and low IN concentration in a cold and dry environment can persist for several days motivates the question of what the major contributors to Arctic mixed-phase cloud dissipation are.

An unique challenge with modeling Arctic mixed-phase clouds is the observed supercooled liquid layer which acts as a direct

link between microphysics and dynamics by cloud-top cooling. This layer forces cloud top cooling which drives the evolution of the cloud by generating a buoyancy-driven vertical overturning (Shupe et al., 2008). New droplets form in the updrafts, while most of the ice nucleates in the cloud layer and grows. The updrafts are limited by the temperature inversion at cloud top. At the same time, cloud-top cooling helps to maintain the inversion (Morrison et al., 2012). Temperature inversions are frequently observed to coincide with humidity inversions in the lower Arctic troposphere (Sedlar and Tjernström, 2009; Devasthale et al.,

2011; Sedlar et al., 2012; Nygård et al., 2014). When temperature inversions coincide with humidity inversions, moisture can potentially be transported downwards into the cloud layer through entrainment near cloud top, a process that acts to maintain the liquid cloud layer (Solomon et al., 2011), and contributes to the observed high relative humidity in the boundary layer (Tjernström et al., 2004; Tjernström, 2005). Depending on whether the cloud is decoupled from the surface or not, moisture sources above cloud top or near the surface are important. Moisture supply helps the mixed-phase cloud to persist in the Arctic

environment and compensate the condensate loss due to formation and fall-out of ice crystals (Zuidema et al., 2005; Morrison et al., 2011).

Simulations of mixed-phase clouds pose additional challenges in models. A high temporal and spatial resolution and a detailed microphysics scheme appear to be essential for representing the boundary layer and the gradients of various parameters, such as liquid water content, properly (Wesslén et al., 2014; de Boer et al., 2014; Solomon et al., 2015; Sotiropoulou et al., 2016).

Several studies have also found that Arctic mixed-phase clouds, and their surface radiative effects, are sensitive to changes in





CCN concentrations. Mauritsen et al. (2011) used observations from the summertime high Arctic to describe a tenuous cloud regime where cloud formation is limited by CCN availability, and where a small increase in aerosols can result in a significant cloud warming effect at the surface. A subsequent modeling study by Birch et al. (2012) confirmed that accurately simulating cloud formation and dissipation under low CCN conditions improves the model representation of the surface energy budget

and temperature.

In situ observations from field campaigns are a key part of improving model simulations of Arctic mixed-phase clouds and their impact on climate. This study exploits observations taken during the Arctic Summer Cloud Ocean Study (ASCOS) in summer 2008 (Tjernström et al., 2012). During this campaign, an Arctic mixed-phase stratiform cloud layer was observed that persisted for an extended period, and then suddenly dissipated.

We seek to find potential mechanisms leading to the dissipation of this cloud layer, using the COnsortium for Small-Scale MOdeling (COSMO) model (Schättler et al., 2015), run in a Large Eddy Simulation (LES) mode with high vertical, horizontal, and temporal resolution, to explore the dissipation. The study is divided into the following sections: Section 2 outlines the ASCOS field campaign and the period of interest. An overview is presented of the model, its setup, and sensitivity experiments in Section 3. Section 4 describes the results of three sensitivity experiments. A discussion and conclusions are presented in

Section 5.

## 2   Overview of the period of interest during ASCOS

The ASCOS campaign took place during summer 2008, and the entire expedition lasted more than one month in the central Arctic Ocean of the North Atlantic (Tjernström et al., 2012). Detailed boundary layer and cloud measurements were taken when the Swedish icebreaker *Oden* drifted with a multi-year ice floe for three weeks around 87° N (Tjernström et al., 2014).

This study will focus on an episode towards the end of the ice drift, around 31 August 2008 (DoY 244). A low-level stratiform cloud layer had been quasi-persistent for about one week but dissipated rapidly in the evening of DoY 244 (Sedlar et al., 2011; Mauritsen et al., 2011; Sotiropoulou et al., 2014). The period of this persistent cloud layer (DoY 236–244) (Fig. 1) was dominated by a high pressure system, with passages of a few weak fronts (Tjernström et al., 2012). A detailed description of the meteorological conditions can be found in Tjernström et al. (2012). Observations from the vertically pointing Doppler

millimeter cloud radar (MMCR) shows the cloud top at around 1 km during the morning hours, with a thinning and lowering cloud top during the afternoon (Fig. 1).

Mixed-phase stratiform clouds often tend to be decoupled from surface layer turbulence by a statically stable layer. During ASCOS, low-level mixed-phased clouds were decoupled from the surface about 75 % of the time (Shupe et al., 2013; Sedlar and Shupe, 2014; Sotiropoulou et al., 2014). The cloud layer shown in Fig. 1 was decoupled during the 8 h period of interest

(Shupe et al., 2013). A CCN counter fed from an inlet on *Oden* approximately 20 m above the surface measured a mean CCN concentration of about $25\,\mathrm{cm^{-3}}$ at a supersaturation of 0.2 % during the time period of the ice drift (DoY 225–246) (Martin et al., 2011). During the evening of DoY 244, CCN concentrations at the surface dropped below $1\,\mathrm{cm^{-3}}$ around the time that





the cloud began to dissipate (Mauritsen et al., 2011; Leck and Svensson, 2015). It is important to understand that since the cloud layer was decoupled, at least initially, we do not know how representative these values are for the cloud layer.

## 3 Model description and setup

The COSMO model was used in a semi-idealized LES setup with periodic boundary conditions similar to Paukert and Hoose (2014). The model domain included 64 grid points in each horizontal direction at a resolution of 100 m. In order to account for the radiative fluxes throughout the atmospheric column (after Ritter and Geleyn, 1992), the top of the model domain was extended up to 22 km, with a vertical resolution of 15 m below 1005 m altitude, and an exponentially decreasing resolution aloft. Subsidence is prescribed as a linear function that increases from zero at the surface to $0.4125\,\mathrm{cm\,s^{-1}}$ at the initialized temperature inversion base (960 m) and remains constant above 960 m. This representation of the subsidence agrees well with ECMWF reanalyses for the 31 August 2008. A two-moment cloud microphysics scheme (Seifert and Beheng, 2006) and a 3-D turbulence scheme (Herzog et al., 2002) were used for the simulations. The surface fluxes, which provide a coupling between the atmosphere and the surface, are related to the turbulence scheme. A TKE (turbulent kinetic energy)-based surface transfer scheme describes the transport through the surface layer (Schättler et al., 2015). Some basic aspects of our simulations follow the model setup of Ovchinnikov et al. (2014), such as fixed number concentrations of both CDNC and ice crystal concentration, large-scale subsidence and a two-hour spin-up period before ice crystal formation. The model surface is set as sea ice, with an albedo of 70 %, consistent with observations. The surface temperature was set to 271.35 K. Changes in surface properties can influence the air-sea interactions and are taken into account by a sea ice scheme (Mironov, 2008). Cloud ice processes were turned off during the initial 2 h of the simulation, in order to permit the liquid cloud layer to develop. The initial temperature and moisture profiles were taken from a radio sounding at 5:35 UTC on 31 August 2008, of which the wind speed and wind direction were smoothed from 12 km to 22 km. The model study was divided into one control simulation and three sets of sensitivity experiments (Tab. 1). The initial profiles were the same in all simulations except for the moisture profiles of sensitivity experiment SensMoist (see below).

### 3.1 Control simulation

The control simulation has a CDNC of $30\,\mathrm{cm^{-3}}$ and a fixed ice crystal concentration of $0.2\,\mathrm{l^{-1}}$; these values are chosen based on the mean values during ASCOS reported in Section 2. Observations at the surface did not record any IN concentrations because they were below the detection limit of the instrument, which ranges between $0.1\,\mathrm{l^{-1}}$ and $2\,\mathrm{l^{-1}}$ (Z. Kanji, personal communication). However, the fact that clouds during ASCOS precipitated predominantly ice crystals implies that IN must have been present (Shupe et al., 2013). It is possible that advection with or without entrainment of IN at cloud top rather than surface sources provided IN for the observed cloud. An earlier field campaign with *Oden* during September 1991 measured a maximum ice-forming nuclei concentration of $0.25\,\mathrm{l^{-1}}$ at 88° N (Bigg, 1996). Guided by these findings, the ice crystal concentration in the model was set to be $0.2\,\mathrm{l^{-1}}$ in the control simulation.



### 3.2 Sensitivity experiments

The first sensitivity experiment (SensMoist) includes 4 simulations where the moisture profile is changed either below the cloud base (sub-cloud layer) or above the cloud top in order to mimic the effect of dry-air advection (Tab. 1). Below cloud, the moisture profile is linearly dried to resemble 99 % relative humidity (RH) at cloud base decreasing to 85 % RH at the surface (Fig. 2, a), while keeping the temperature profile the same as in the control simulation. Above cloud top, a 450 m deep layer of the atmosphere is progressively dried in 3 different simulations corresponding to RH values of 36 %, 20 % and finally 10 % above and in contact with cloud top (Fig. 2, b-d).

In order to investigate the sensitivity of the modeled cloud to changes in ice crystal concentrations, the ice crystal concentration was increased to values well above the expected low values in the Arctic in the second set of sensitivity experiments (SensIce). Two simulations with ice crystal concentrations set to $1\,l^{-1}$ and $10\,l^{-1}$ were conducted.

The third sensitivity experiment (SensCDNC) considers the low CCN concentrations observed during the ASCOS field campaign. During DoY 244, CCN concentrations at the surface dropped below $1\,cm^{-3}$ (Mauritsen et al., 2011; Leck and Svensson, 2015). The CDNC was decreased to $2\,cm^{-3}$ and $10\,cm^{-3}$, respectively, in two simulations to investigate the impact of low CDNC on the mixed-phase cloud development.

## 4 Results

### 4.1 Control simulation

The initial $\theta$-profile shows a neutral to stable boundary layer (Fig. 3). A small inversion is seen in both $\theta$ and $\theta_e$ profiles near 300 m; this is the decoupling inversion, separating turbulence driven by the cloud layer from surface-driven turbulence. After 4 h of simulation, the model $\theta_e$-profiles display a well-mixed layer extending from the surface to the inversion base near 1000 m; this is also where the cloud top is located (Fig. 4). The boundary layer deepens over the next 8 h, causing the main inversion and the cloud top to rise by around 90 m (Fig. 4). The $\theta$ profiles imply that the lower half of the boundary layer transitions towards less stable and hence the decoupling inversion around 300 m disappears after 4 h. The boundary and cloud layers thus quickly become coupled in the simulations. This tendency to erode cloud decoupling is common in LES simulations (Savre et al., 2014).

The maximum ice water content (IWC) in the control simulation is two orders of magnitude smaller than the liquid water content (LWC), around $0.0015\,g\,kg^{-1}$ (Fig. 4). After 3 h the mixed-phase cloud generates ice and begins precipitating ice crystals which fall through the sub-cloud layer and reach the surface (Fig. 4, red). At the same time a secondary cloud layer briefly forms at the decoupling inversion, likely associated with a moistening of the sub-cloud layer through ice crystal sublimation. Rain (rain water content (RWC)) precipitates out of the liquid layer after around 7 h but does not reach the surface due to evaporation and conversion to ice (Fig. 4, green). When a cloud droplet grows to a diameter of 80 $\mu$m, it is defined as a rain droplet in the model. After 4 h, when ice formation is relatively constant, the control simulation develops a liquid cloudy layer





that is persistent throughout the simulation with a thickness of approximately 200 m (Fig. 4, blue). The maximum LWC in the cloud is around $0.2\,\mathrm{g\,kg^{-1}}$.

## 4.2 Sensitivity experiment - SensMoist

The availability of moisture above and below the cloud is an important ingredient for the persistence of an Arctic mixed-phase
cloud. Fig. 5 shows the evolution of cloud liquid water path (LWP) in the SensMoist experiment (pink lines). Reducing the available moisture in the atmosphere below the cloud does not change the persistence of the cloud. Up to 8 h of simulation, the LWP is slightly smaller (approximately $8\,\mathrm{g\,m^{-2}}$ after 4 h) than in the control simulation and at the end of the simulation, the LWP is almost the same (Fig. 5, pink solid line). This suggests that the supply of moisture from near the surface has only a limited influence on the cloud layer, resulting in a stable LWP around $50\,\mathrm{g\,m^{-2}}$. Imposing a region of dry-air above the cloud has a larger influence on the cloud evolution. Drier air above the cloud layer leads to a decrease in LWP in all three simulations (Fig. 5, dashed pink lines). The reduction is strongest when RH above the cloud was reduced to $10\,\%$. The LWC is reduced by almost a factor of 2 compared to the control simulation (Fig. 6). When the source of moisture from above is decreased, the boundary layer and cloud layer become coupled between 2 h and 4 h, which is similar to the control simulation (Fig. 7).

The $\theta_e$ profiles also show a clear weakening of the inversion after 2 h which is due to the thinner cloud layer and consequently decreased turbulence (Fig. 7). Hence the boundary layer cannot grow with time as it does in the control simulation (Fig. 4, 6). Following the reduction in LWC, IWC is also reduced relative to the control simulation; the mass of the liquid droplets is decreased and therefore ice crystals grow less rapidly. This causes the ice crystals to remain suspended in the atmosphere longer due to their reduced size and fall speed (6, red). These results examining the sensitivity of cloud to the moisture profile changes agree with the behavior of the Arctic mixed-phase cloud as reported in Solomon et al. (2013).

## 4.3 Sensitivity experiment - SensIce

In the simulation with an increased ice crystal concentration to $1\,\mathrm{l^{-1}}$, the cloud still persists over the simulation time, and IWC increases because of the large number of ice crystals (Fig. 8). The impact on the liquid layer is however marginal. The LWP is almost constant at around $50\,\mathrm{g\,m^{-2}}$, very similar to the LWP evolution simulated when RH is reduced in the sub-cloud layer (Fig. 5). Further increasing the ice crystal number to $10\,\mathrm{l^{-1}}$ leads to glaciation and finally to dissipation of the cloud after 6 h (Fig. 5, blue dashed line). In this simulation, the inversion near cloud top becomes weaker after 8 h and the weak stable layer near 300 m erodes more rapidly than in the control simulation (Fig. 9).

## 4.4 Sensitivity experiment - SensCDNC

When CDNC is reduced relative to the reference value in the control simulation, the LWP time series shows a decrease to around $40\,\mathrm{g\,m^{-2}}$ with the CDNC $10\,\mathrm{cm^{-3}}$, and to below $10\,\mathrm{g\,m^{-2}}$ for CDNC set to $2\,\mathrm{cm^{-3}}$ (Fig. 5, black lines). The reduction in CDNC also leads to a weakening of the inversion around 1 km, while the inversion near 300 m persists throughout the simulation duration, whereas it is eroded after roughly 4 h in the control simulation (Fig. 10). The weakening of the main





inversion is likely due to less radiative cooling at the cloud top, because of the optically thinner, less opaque liquid layer; this also decreases the cloud overturning circulation which in turn slightly strengthens the decoupling inversion. With an optically thinner cloud above, the sub-cloud layer can cool more efficiently, and this promotes the formation of secondary, thin liquid layer in the vicinity of the lower temperature inversion near 300 m (Fig. 11). Rain forms after 2 h from initialization, through

collision and coalescence processes. Rain from the main cloud layer can moisten the sub-cloud layer due to evaporation until the cloud layer at 1 km almost dissipates. This simulation leads to a very thin cloud with LWC values reaching $0.03\,\mathrm{g\,kg^{-1}}$ and maximum values of IWC of $0.0015\,\mathrm{g\,kg^{-1}}$ close to the surface. Ice crystals falling from the upper cloud layer pass through the lower liquid layer around 3 h simulation time, where they grow at the expense of cloud droplets, resulting in IWCs as large as the control simulation. This also causes the second, lower liquid cloud to become tenuous and briefly intermittent (Fig. 11).

## 5   Discussion and conclusions

Low aerosol concentrations are common in the high Arctic due to a lack of aerosol sources in this region in particular during summer (Bigg, 1996; Heintzenberg et al., 2006; Garrett et al., 2010; Heintzenberg and Leck, 2012). In persistent precipitating boundary layer clouds, the aerosol concentration can be further reduced through scavenging. Thus, changes in aerosol concentrations and consequently CDNCs may strongly influence the lifetime and development of an Arctic mixed-phase cloud. Our current model study of an observed mixed-phase cloud during the ASCOS field campaign shows that a CDNC concentration of $10\,\mathrm{cm^{-3}}$ is sufficient to sustain the cloud while a CDNC of $2\,\mathrm{cm^{-3}}$ leads to cloud dissipation.

The results are in agreement with Mauritsen et al. (2011), who discussed a tenuous cloud regime in the Arctic characterized by low CCN number concentrations. Mauritsen et al. (2011) found that a CCN number concentration of $10\,\mathrm{cm^{-3}}$ marked the upper boundary for a transition regime below which cloud formation becomes limited. Observations during three previous campaigns in the high Arctic and during the ASCOS field campaign indicate a 25 to 30 % occurrence frequency of CCN concentrations below this value, i.e. within the so-called tenuous cloud regime (Mauritsen et al., 2011); during ASCOS the median CCN concentration was 20 to $30\,\mathrm{cm^{-3}}$ as measured by two independent CCN counters set to the same supersaturation; CCN concentrations were below $10\,\mathrm{cm^{-3}}$ about 20 to 30 % of the time (Tjernström et al., 2014).

Using both a 3D and a single-column version (SCM) of the MetUM numerical weather prediction model, and exploring an extended period of ASCOS observations, Birch et al. (2012) found that a constant CCN concentration of $10\,\mathrm{cm^{-3}}$ instead of $100\,\mathrm{cm^{-3}}$ gave a better general representation of low-level mixed-phase cloud properties. In a study of Arctic stratocumulus clouds and dynamic surface coupling, Sotiropoulou et al. (2014) used an indirect method to show that that the presence of optically thin clouds observed during ASCOS correlate with low CCN concentrations and persist for about 30 % of the time. The analysis by Mauritsen et al. (2011), Birch et al. (2012), and the findings in the present study indicate that a drop in aerosol and CCN number concentration to values below $10\,\mathrm{cm^{-3}}$ may be an important reason for mixed-phase cloud dissipation in the high Arctic in summer. It is therefore important that models, in particular with interactive aerosol and cloud microphysics, can represent this type of low aerosol cloud regime, while many models assume constant droplet number or aerosol concentrations representative for mid-latitudes (Wesslén et al., 2014; Sotiropoulou et al., 2016).





While Birch et al. (2012) ran their simulations over several days, the COSMO simulations presented here are for 10 h and focused on the cloud development during that time. With the high horizontal and vertical resolution, the COSMO simulations focus on the cloud microphysics and cloud evolution over a shorter time compared to the MetUM SCM simulations of Birch et al. (2012). The idealized setup with periodic boundary conditions and the small domain limit the investigation area and hence

focus only on parts of the Arctic stratus cloud deck.

The COSMO simulated cloud was also sensitive to changes in the moisture profile. Generally, LWC decreased when RH in the atmospheric layer above the cloud top was decreased. This supports observational and modeling evidence suggesting that the source of water vapor above cloud top is important for the persistence of the liquid layer (Solomon et al., 2011; Sedlar et al., 2012; Morrison et al., 2012). However, in our simulations, introducing a dry layer above the inversion did not cause cloud

dissipation. Reducing RH in the sub-cloud layer had only a modest impact on the mixed-phase cloud. Mixed-phase clouds in the Arctic are frequently decoupled from the surface (Sedlar and Shupe, 2014; Sotiropoulou et al., 2014) and therefore do not necessarily rely on a moisture source near the surface to persist.

Increasing the ice crystal concentration to $1\,l^{-1}$ had a moderate influence on the simulated mixed-phase cloud, while an even higher ice crystal concentration of $10\,l^{-1}$ led to glaciation and subsequent dissipation of the cloud. Rogers et al. (2001) found

that for thin, low-level stratus clouds, the IN concentration at -15 to -20 °C was around $1\,l^{-1}$. Nevertheless, ice crystal concentrations in the Arctic may vary over 3 orders of magnitude (Morrison et al., 2005) and a maximum ice crystal concentration of $0.25\,l^{-1}$ has been observed in a similar season and geographic region as ASCOS (Bigg, 1996). Considering IN concentration of $0.25\,l^{-1}$ or lower from a past field campaign in the high Arctic (Bigg, 1996) and taking the absence of IN measurements above the instrument detection limit during ASCOS into account, such a large increase in ice crystal number concentration,

seems an unlikely mechanism responsible for the observed cloud dissipation during ASCOS. Hence, these results suggest that reasonable increases in IN concentrations are not the primary mechanism leading to cloud dissipation for this observed case.

The sensitivity experiments tested here, altering CDNC, ice crystal number concentration, and changing moisture sources to the cloud layer, were designed to mimic changes in the large-scale circulation and advection of air masses with different thermodynamic profiles and aerosol properties. In reality, it is likely that changes in thermodynamical properties and aerosol will

happen simultaneously, and that the combination of these processes will control the evolution of the mixed-phase cloud. Nevertheless, we have shown that, independently, dry-air advection above cloud top, ice crystal increase, and CDNC reduction all contribute to a reduction of the liquid condensate layer of a mixed-phased cloud. However, we find that the reduction of CDNC was likely the primary contributor to the dissipation of the observed mixed-phase cloud during this specific case.

*Acknowledgements.*   K. Loewe was funded by the Helmholtz GRAduate school for Climate and Environment (GRACE, www.GRACE.kit.edu)

during her stay at Stockholm University. C. Hoose acknowledges funding by the Helmholtz Association through the Climate Initiative REKLIM and the President's Initiative and Network Fund (VH-NG-620).




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





**Table 1.** Overview of different sensitivity simulations

| Simulation | Specifications |
|---|---|
| Control simulation | CDNC = $30\,\text{cm}^{-3}$, ice crystal concentration = $0.2\,\text{l}^{-1}$ |
| Sensitivity experiment 1 (SensMoist) | dry-air advection below the cloud |
| | dry-air advection above the cloud with RH of $36\,\%$ |
| | dry-air advection above the cloud with RH of $20\,\%$ |
| | dry-air advection above the cloud with RH of $10\,\%$ |
| Sensitivity experiment 2 (SensIce) | ice crystal concentration = $1\,\text{l}^{-1}$ |
| | ice crystal concentration = $10\,\text{l}^{-1}$ |
| Sensitivity experiment 3 (SensCDNC) | CDNC = $10\,\text{cm}^{-3}$ |
| | CDNC = $2\,\text{cm}^{-3}$ |

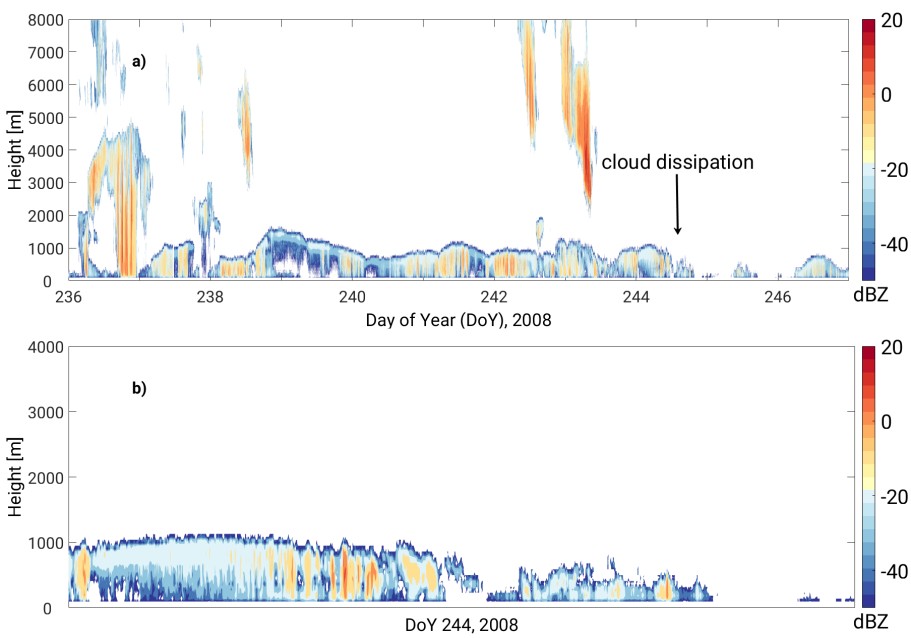

**Figure 1.** Radar reflectivity factor (colours, dBZ) contour time series for the period DoY 236 to 246 during 2008 **(a)** and for DoY 244 **(b)**.



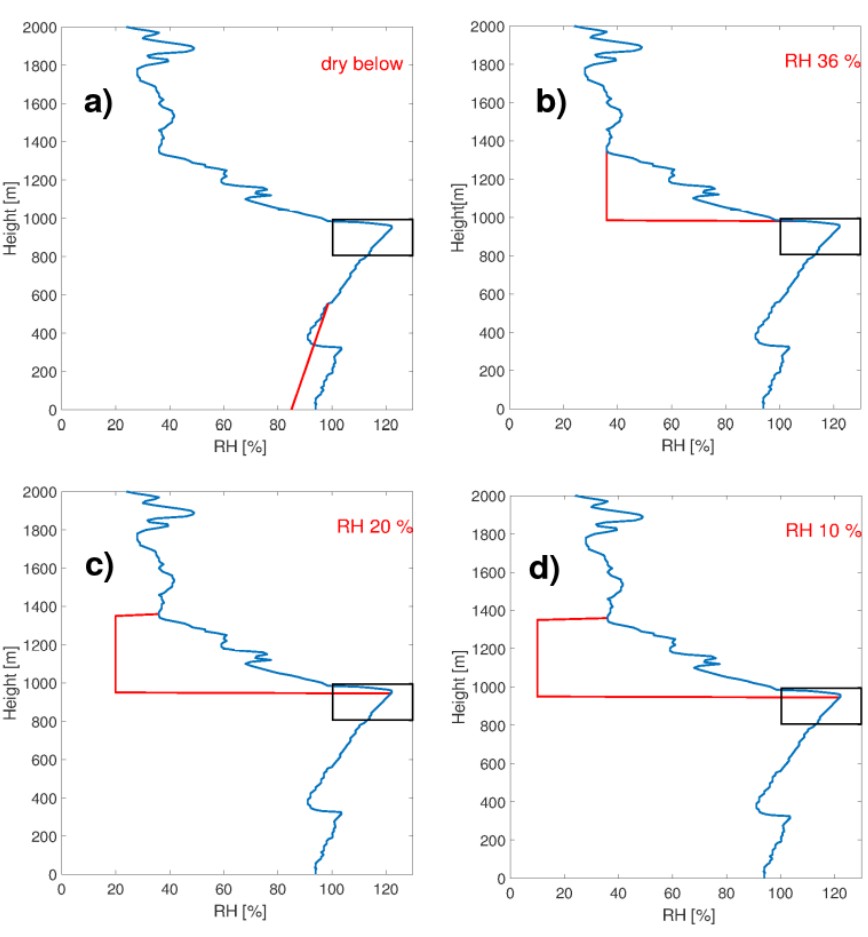

**Figure 2.** RH profiles for the sensitivity experiment SensMoist simulations. Modified parts of the RH profile are shown in red. The initial RH profile is in blue. **(a)** RH profile for the simulation with dry-air advection below the cloud. **(b)** shows the RH profile of the simulation with the dry-air advection above the cloud with the RH of 36 %, **(c)** and **(d)** show simualtions with the dry-air advection above the cloud with the RH of 20 % and 10 %, respectively. Black box marks the vertical extent of the liquid cloud layer of the control simulation.





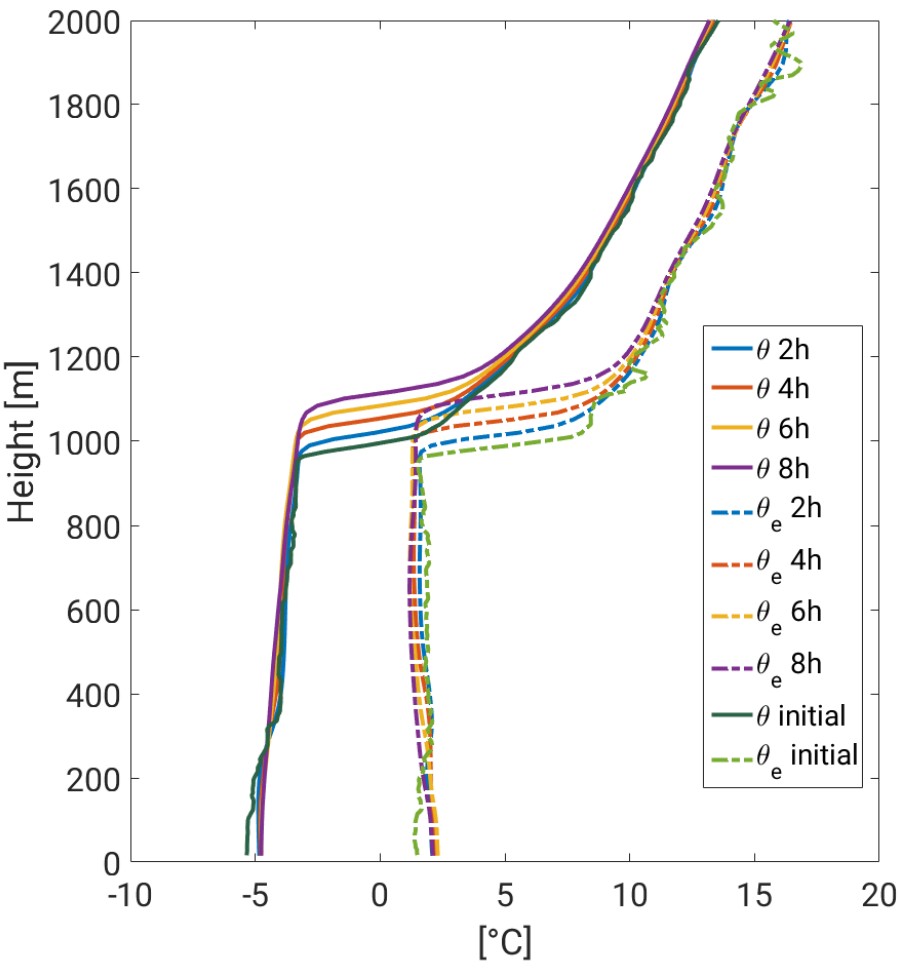

**Figure 3.** $\theta$ (left) and $\theta_e$ (right) profiles of the initial conditions (dark green and green), and after 2 h (blue), 4 h (orange), 6 h (yellow) and 8 h (purple) from the start of the control simulation.





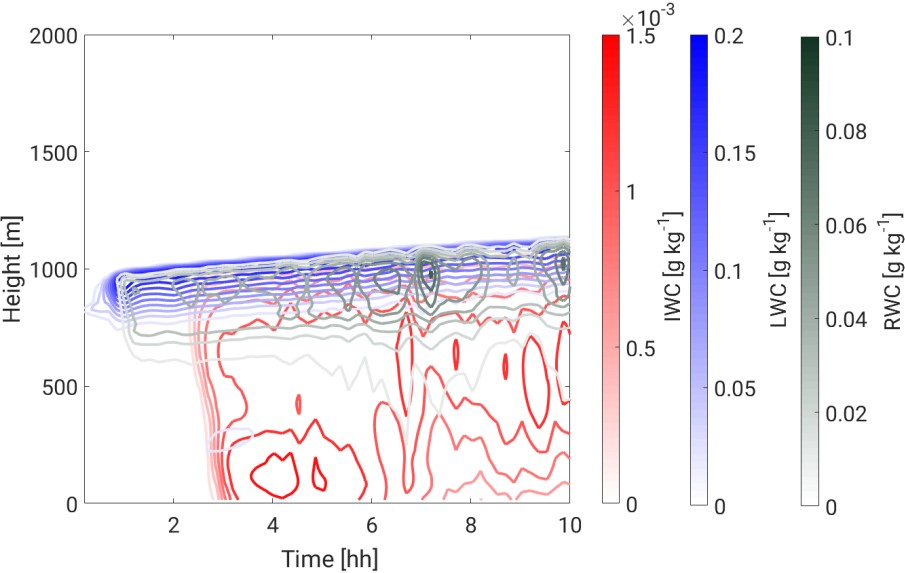

**Figure 4.** Mean values of LWC (blue), IWC (red) and RWC (green) for the control simulation.

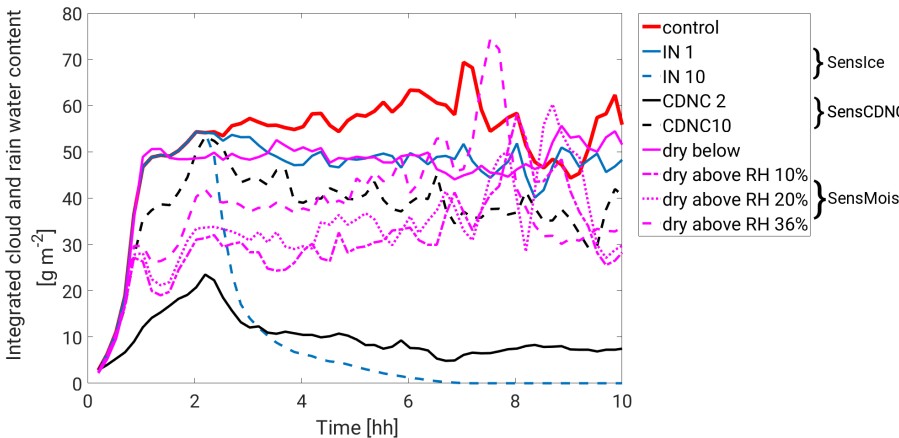

**Figure 5.** The domain-averaged LWP (including cloud droplets and rain) for the control simulation (red), the simulation of dry-air advection below the cloud (pink solid), the simulations of dry-air advection above the cloud top with a RH of 36 % (pink dashed), a RH of 20 % (pink dotted) and a RH of 10 % (pink dash-dotted), the simulations with an ice crystal concentration of $1 l^{-1}$ (blue solid) and $10 l^{-1}$ (blue dashed), the simulation with a CDNC of $10 \, cm^{-3}$ (black dashed) and a CDNC of $2 \, cm^{-3}$ (black solid).





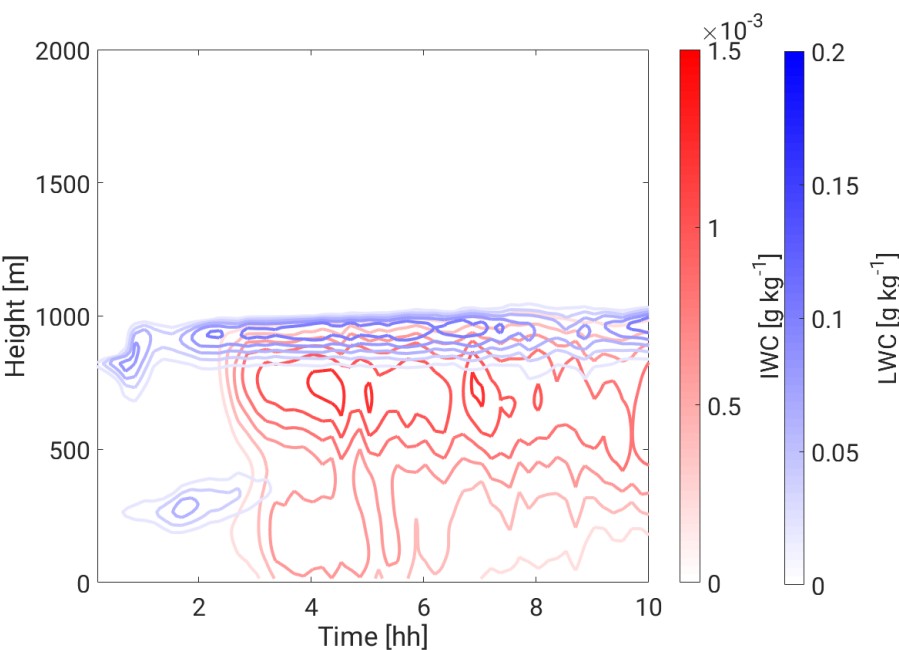

**Figure 6.** Mean values of LWC (blue) and IWC (red) for the SensMoist simulation with the RH of 10 % above the cloud top.





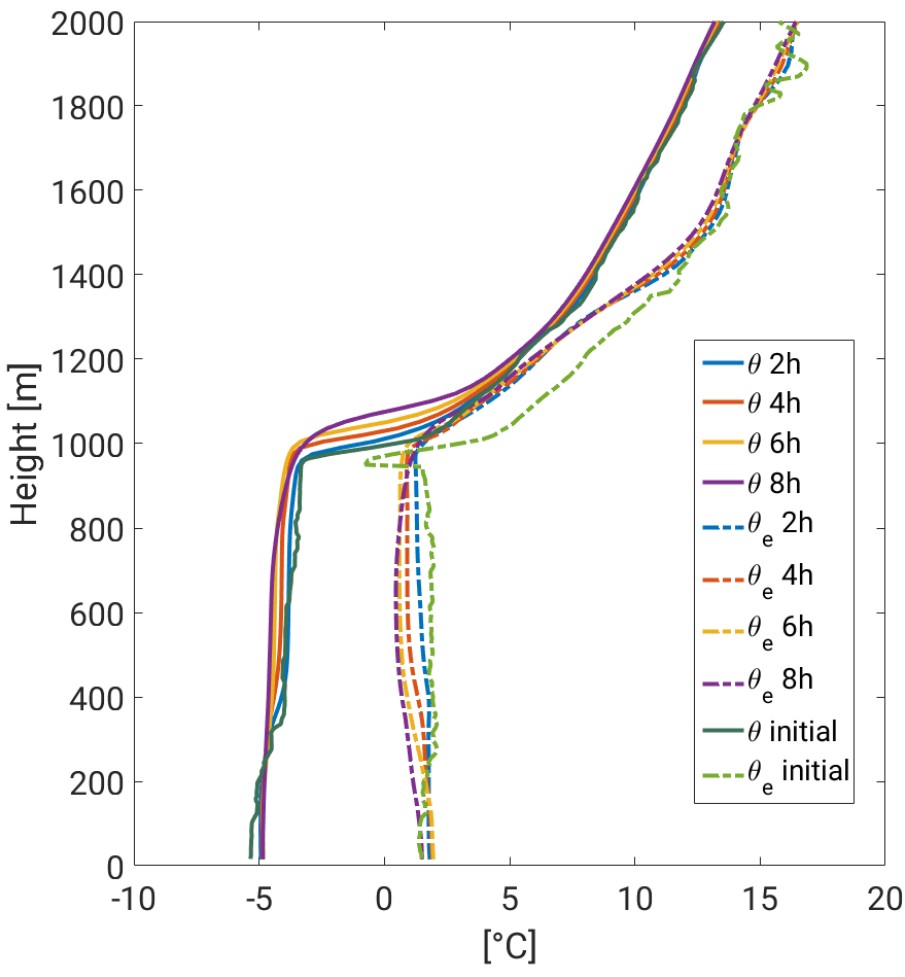

**Figure 7.** $\theta$ (left) and $\theta_e$ (right) profiles of the initial conditions (dark green and green) and after 2 h (blue), 4 h (orange), 6 h (yellow) and 8 h (purple) after the start of the SensMoist simulation with a RH of 10 %.



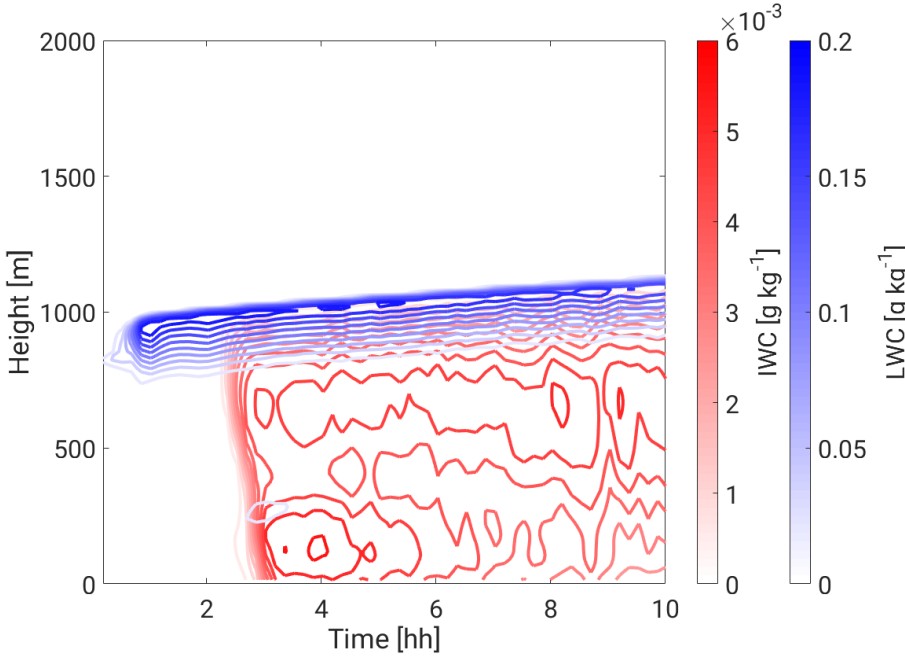

**Figure 8.** Mean values of LWC (blue) and IWC (red) of the SensIce simulation with an ice crystal concentration value of $1\,1^{-1}$.





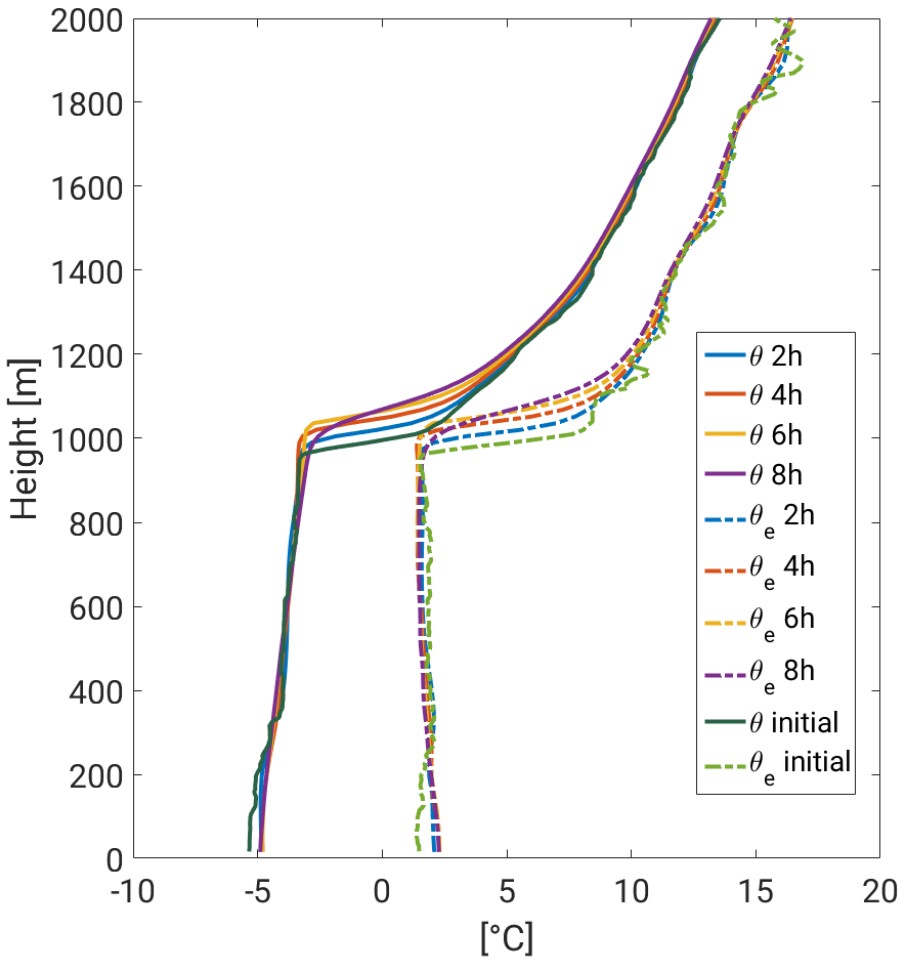

**Figure 9.** $\theta$ (left) and $\theta_e$ (right) profiles of the initial conditions (dark green and green) and after 2 h (blue), 4 h (orange), 6 h (yellow) and 8 h (purple) after the start of the SensIce simulation with an ice crystal concentration $101^{-1}$.





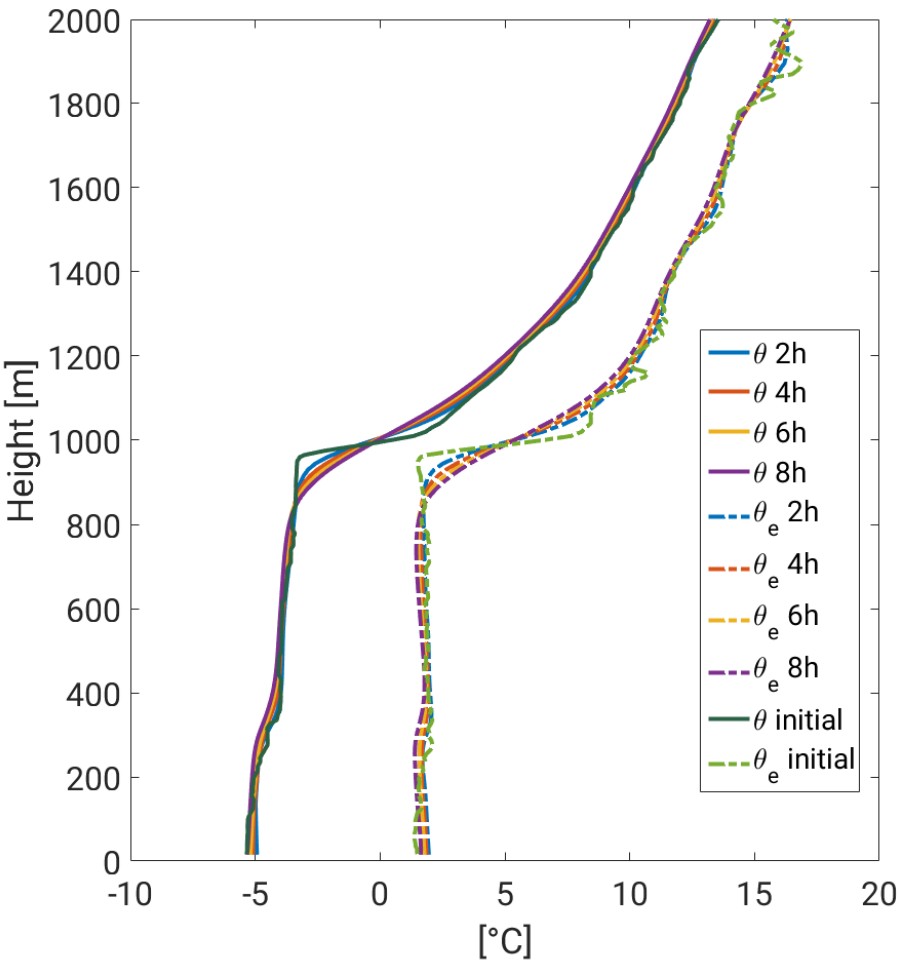

**Figure 10.** $\theta$ (left) and $\theta_e$ (right) profiles of the initial conditions (dark green and green) and after 2 h (blue), 4 h (orange), 6 h (yellow) and 8 h (purple) after the start of the SensCDNC simulation with a decreased CDNC of $2\,\mathrm{cm}^{-3}$.





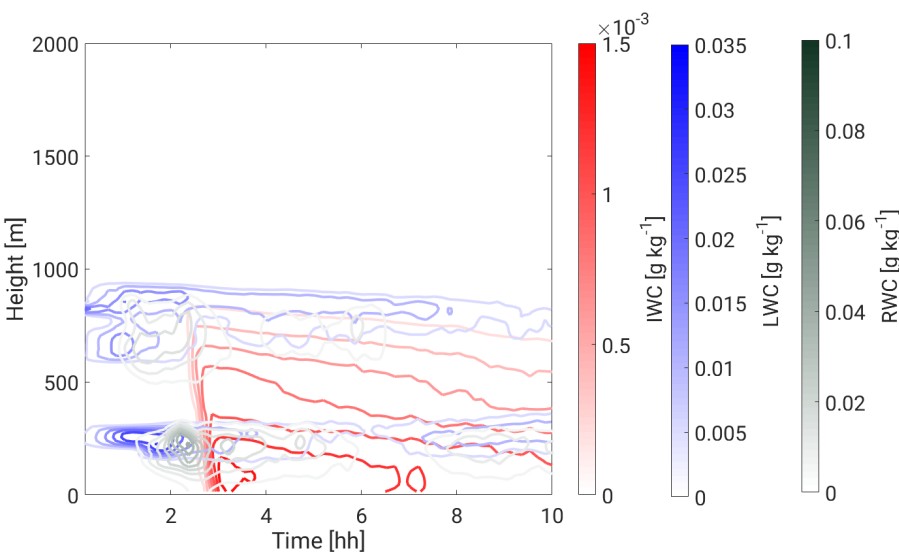

**Figure 11.** Mean values of LWC (blue), IWC (red) and RWC (green) of the SensCDNC simulation with $2\,\text{cm}^{-3}$ CDNC.