# Peer review of "Modelling micro- and macrophysical contributors to the dissipation of an Arctic mixed-phase cloud during the Arctic Summer Cloud Ocean Study (ASCOS)"

_Atmospheric Chemistry and Physics, 2016_

## Referee Comment (RC1) · Anonymous Referee #3 · 4 Jan 2017

General comment:

This paper demonstrates impacts of dry air advection, cloud droplet number concentration, and ice crystal number concentration on dissipation of Arctic mixed-phase clouds. The results clarified that increase in cloud droplet number concentration, resulting from increase in aerosol concentration, would be more important for dissipating process of mixed-phase clouds. The results obtained in this study would be of value in understanding formation and dissipation processes of Arctic mixed-phase clouds. However, this study seems to lack evaluation of model results. I recommend describing how well

the simulations represented realistic mixed-phase clouds over the Arctic in revising version.

Major comments:

1. How was this observed case identified as mixed phase? The case in this study was selected from the summer season, and temperatures through the cloud would not be below the freezing level enough. In fact, the simulated cloud by control run produced rain. So, I suspect that temperatures within the cloud could be partly above the freezing level, and the cloud droplets were not supercooled. Please show evidences from observations that the selected case certainly contained mixed-phase cloud (e.g., lidar, MWR, and ceilometer measurements).

2. How well did the model simulate the observed mixed-phase cloud? Did the simulated clouds well simulate the observation? Please evaluate the simulations and show how the simulated results represented the observation in terms of, for example, LWP (LWC/IWC), cloud base/top heights, horizontal distribution of clouds, ice particle number concentration, etc. The control simulation produced significant amount of rain. -Is this realistic? The evaluations of those simulations would help to make the results from sensitivity experiments more robust.

3. How representative is the chosen case of Arctic mixed-phase clouds over the Arctic? Are the clarified mechanisms unique in the Arctic region, or can they be extended to apply to other environments such as mid latitude?

Minor comments:

1. P. 4, line 20, "smoothed from 12 km to 22 km": Are these numbers meaning altitudes? 2. Figures 3, 7, 9, and 10: Did those profiles indicate model domain average? 3. P. 5, lines 21-22, "The $\theta$ profiles imply that ...": I cannot see this in Fig. 4. It seems to me that $\theta$e showed this characteristics. 4. P. 6, line 7, "the LWP is slightly smaller (approximately 8 g m^-2 after 4 h)": I suppose that this sentence mentioned the CDNC

2 simulation. If so, I think that a value of "8 g mˆ-2" in LWP is significantly smaller than that from the control simulation; it does not look like "slightly smaller". 5. I recommend referring a Heike Kalesse's (2016) paper titled "Understanding rapid changes in phase partitioning between cloud liquid and ice in stratiform mixed-phase clouds: An Arctic Case Study". This paper also mentioned a dissipation mechanism of Arctic mixed-phase cloud observed at Barrow, AK, on the basis of observation and partly model analysis. (Kalesse et al. 2016, Mon. Wea. Rev., doi: http://dx.doi.org/10.1175/MWR-D-16-0155.1)

---

## Referee Comment (RC2) · Anonymous Referee #2 · 13 Feb 2017

In the present study the authors attempt to identify the potential mechanisms leading to the dissipation of a mixed-phase boundary-layer cloud in the high Arctic. The model simulations presented in the paper are based on an observed case from the recent ASCOS field campaign. The authors examine several processes that might be contributing to the cloud layer dissipation and conclude that the most likely reason is the reduction of cloud droplet concentration as a result of diminished background aerosols.

The paper is very well written and easy to read. At the same time, while I do appreciate the authors' effort to keep the paper short and up to the point, I think that it would benefit from expanding the analysis of the simulations and including some additional figures.

I would agree with Reviewer #3 that the lack of observational verification is a major limitation of this study. Although the authors specifically state that their study exploits the observations collected during ASCOS, there is only a radar reflectivity plot and a moisture sounding presented in the paper. I doubt detailed in-situ microphysical data are available for this case but perhaps some observations or reanalysis results characterizing the large-scale environment, which to a large extent determines the evolution of the cloud layer, could be included.

Overall, I would recommend publication after the following comments are addressed in the revised version.

Major comments:

1. Since the CCN-limited regime is the main focus of the paper, I would expect that all simulations would be carried out using prognostic CCN instead of fixed CDNC. I consider this a major flaw that obviously warrants some discussion in the paper. What are the potential implications and limitations of this approach? For example, I would assume that with prognostic CCN the cloud layer would start diminishing at much higher background aerosol concentrations.

2. I'm not sure I understand the rationale behind the SensMoist sensitivity simulations. If the intent was to mimic a large-scale drying advection, then why modify the initial moisture profile and not just impose a forcing term that would dry the domain out throughout the simulation? Changing the initial moisture profile doesn't seem to be a very realistic representation of drying large-scale advection. Is the magnitude of this drying consistent with ECMWF reanalysis or following ASCOS soundings?

Also, how realistic is it to impose drying advection either only below or only above the cloud layer? Obviously, there are two sources of moisture in the simulations – one above the cloud layer and one below; therefore turning them off one by one does not seem to be an effective way of turning the moisture supply off. Speaking of moisture sources, what are the surface fluxes in the simulations?

I would also strongly suggest including the vertical $q_v$ and moisture turbulent flux profiles when discussing the simulations.

3. The analysis of SensCDNC simulations seems a bit dry. I would suggest expanding it, as this is the main point of the paper. Without knowing the simulations in detail, I could speculate that the effect of decreasing CDNC is most likely "indirect", i.e. lower CDNC leads to lower LWC, and consequently less radiative cooling, weaker cloud layer circulation that cannot penetrate and mix the inversion at 300m, thus preserving the initial two-layer structure. In this configuration, the top cloud layer is isolated from the moisture source below, the moisture source above is weakened by the decreased entrainment and the cloud layer begins to dissipate. The bottom layer, however, seems quite robust and does not show any signs of decay, which obviously contradicts the observations in Fig. 1.

Minor comments:
1. Pg. 4 ln. 5: I assume that the horizontal grid spacing is 100m, not the model resolution. Correct? The same on ln. 7 for the vertical grid spacing.

2. What is the ice particle habit used in the simulations? Is it consistent with observations and/or temperature regime?

3. Pg. 5 ln 30: "When a cloud droplet grows …" – perhaps the model description section would be a better place for this sentence?

---

## Author Comment (AC1)

**Statement on the Revision of acp-2016-917 Based on the Referees' Report**

Katharina Loewe    Annica M. L. Ekman    Marco Paukert
Joseph Sedlar    Michael Tjernström    Corinna Hoose

March 26, 2017

This document contains detailed responses to the 2 reviewers' comments and criticisms of our manuscript titled: " Modelling micro- and macrophysical contributors to the dissipation of an Arctic mixed-phase cloud during the Arctic Summer Cloud Ocean Study (ASCOS)", for potential publication in Atmos. Chem. and Phys. (acp-2016-917). We thank the reviewers for their critical comments, especially regarding their concern for a more thorough comparison with ASCOS observations. Below, we list the specific review comments/concerns (marked with a gray side bar), followed by our responses. Notice that figure numbers (may) have changed from the original submission due to the additional figures included in the revised manuscript.

**Comments by Reviewer #3**

> General comment:
> This paper demonstrates impacts of dry air advection,
> cloud droplet number concentration, and ice crystal number
> concentration on dissipation of Arctic mixed-phase clouds.
> The results clarified that increase in cloud droplet
> number concentration, resulting from increase in aerosol
> concentration, would be more important for dissipating
> process of mixed-phase clouds.  The results obtained in
> this study would be of value in understanding formation
> and dissipation processes of Arctic mixed-phase clouds.
> However,this study seems to lack evaluation of model
> results.  I recommend describing how well the simulations
> represented realistic mixed-phase clouds over the Arctic in
> revising version.

Thank you for your comments. We have made a more extensive evaluation of the model simulations against observations performed during ASCOS. Please see more detailed responses to individual comments below.

Major comment 1:
How was this observed case identified as mixed phase?  The
case in this study was selected from the summer season,
and temperatures through the cloud would not be below the
freezing level enough.  In fact, the simulated cloud by
control run produced rain.  So, I suspect that temperatures
within the cloud could be partly above the freezing level,
and the cloud droplets were not supercooled.  Please
show evidences from observations that the selected case
certainly contained mixed-phase cloud (e.g., lidar, MWR, and
ceilometer measurements).

The fact that these observations occurred during the summer, more so during late summer season, does not mean the lower tropospheric temperatures are not cold enough to support ice crystal formation. During the summer over the central Arctic ice pack, clouds are generally found to consist of a combination of supercooled liquid and ice crystals (e.g., [9]). To further address cloud phase, we have analyzed the ASCOS cloud phase classification data product. This product provides a time-height dataset of cloud and precipitation hydrometeor classification. It is produced by combining multiple remote sensing and in-situ measurements; these include the millimeter cloud radar (MMCR), dual channel microwave radiometer (MWR), ceilometer, and 6-hr radiosounding profiles. The methodology follows that of [8]. At a height of $725\,$m and $950\,$m the cloud is classified as "mixed-phase" for most of the first $10\,$h of the 31 August 2008 with intermittent classifications as "liquid" or "ice". (Fig. 2, c, d). Thus, we can say the cloud classification was during the first half of the 31 August 2008 was mixed-phase. Furthermore, the cloud top temperature observed by a radiosonde was around -10 C° and thus the cloud droplets were supercooled (Fig. 1, here in the reply text). We added two sentences and two figures in section 2 of the manuscript to identify the cloud type: "With observations from the MMCR, a dual-channel microwave radiometer (MWR), a ceilometer, and radiosondes the cloud type was classified as mixed-phase during the first half of the 31 August 2008. The cloud type classification follows the method by [8] (Fig. 2, c, d)."

[Figure]

Figure 1: Radiosonde profile on 31 August 2008 at 5:35UTC.

```
Major comment 2:
How well did the model simulate the observed mixed-phase
cloud?  Did the simulated clouds well simulate the
observation?  Please evaluate the simulations and show
how the simulated results represented the observation
in terms of, for example, LWP (LWC/IWC), cloud base/top
heights, horizontal distribution of clouds, ice particle
number concentration, etc.  The control simulation produced
significant amount of rain.  -Is this realistic?  The
evaluations of those simulations would help to make the
results from sensitivity experiments more robust.
```

To answer this comment, we have added the cloud base height measured by the ceilometer to Fig. 1 and compared it in section 4.1 with the simulated height. "The simulated cloud top corresponds well with the cloud top seen by the MMCR at around 1 km (Fig. 1, b). The cloud base, measured with a laser ceilometer, is between 600 m and 700 m at the beginning of DoY 244. This altitude agrees well with the cloud base height of the simulated cloud layer, which is around 600 m (Fig. 5 , previously Fig. 4)."

Furthermore, we have compared the observed LWP and IWP with the control simulation. We also discuss the uncertainty in the observed LWP and IWP, because these should be taken into account when compared with the model.We add a short paragraph in section 2 about the uncertainty and compared the LWP and IWP in section 4.1. We have added the following sentences in section 2: "The laser ceilometer measured the cloud base at around 600 m to 700 m in the morning. During the day, the cloud base decreased towards the surface. With observations from the MMCR, a dual-channel microwave radiometer (MWR), a ceilometer, and radiosondes, the cloud type was classified as mixed-phase during the first half of the 31 August 2008. The cloud type classification follows the method by [8]. The retrieval of the liquid water path (LWP) from the MWR contains of an uncertainty of 25 g m-2 [11], explaining the negative LWP observations in Fig. 2a. During DoY 244, the LWP increased from around $90 \, \mathrm{g \, m^{-2}}$ to values over $300 \, \mathrm{g \, m^{-2}}$ and varied considerably during the first half of the day. Finally, the LWP reached values around $50 \, \mathrm{g \, m^{-2}}$ in the afternoon. The ice water path (IWP) is integrated from profiles of the ice water content (IWC), which are derived from MMCR reflectivity power-law relationships at vertical levels deemed to predominantly ice-phase by the cloud phase classifier [7, 6]. The uncertainty in IWC retrieval, as large as a factor of two [7, 6] results from a combination of systematic and random errors. The IWP was in the range of $10 \, \mathrm{g \, m^{-2}}$ in the morning and varied over a wide range until 12 UTC. After 12 UTC the IWP was around or even below $5 \, \mathrm{g \, m^{-2}}$ (Fig. 2)."

Furthermore, we have added a comparison of the simulated and observed LWP and IWP in section 4.1: "Observations of the LWP show a more variable LWP in the morning than in the afternoon (Fig. 2, a, b). The simulated LWP is around $50 \, \mathrm{g \, m^{-2}}$ and most of the time in the range of the observed LWP, which has

an uncertainty of $25\,\mathrm{g\,m^{-2}}$. Because the simulated cloud is not dissipating in the control simulation, the simulated LWP remains in that range and is not decreasing during the day. The IWP of the control simulation seems to be at the lower end of the observed IWP range and reaches only around $2\,\mathrm{g\,m^{-2}}$ after the ice processes are turned on."

We see in Fig. 5 (previously Fig. 4) that the COSMO model produces rain, while there is almost no precipitation observed reaching the surface. Fig. 1b show that hydrometeors are falling out out of the cloud. Within the model a cloud droplet is classified as a raindrop when a certain size is reached. A raindrop has a minimum size of $80\,\mu m$. The rain water content (RWC) in Fig. 5 (past 4) shows, that rain is produced in the cloud layer and does not reach the ground. Hence, the precipitation amount at the surface is small. However, the COSMO model seems to have a very effective autoconversion rate of cloud droplets to raindrops. This behavior is also recognized in the model intercomparison study of the BACCHUS project (www.bacchus-env.eu/), where the COSMO model participates. This study will be submitted later in 2017 and will help to quantify the different microphysical processes in Arctic mixed-phase clouds in different models (Stevens et al., 2017, in prep.).

> Major comment 3:
> How representative is the chosen case of Arctic mixed-phase
> clouds over the Arctic? Are the clarified mechanisms unique
> in the Arctic region, or can they be extended to apply to
> other environments such as mid latitude?

The Arctic is a very remote region and these low CCN concentrations are not common at mid latitudes, because of the more abundant natural and anthropogenic emission sources. Thus, one can say that the mechanism of dissipation of an Arctic mixed-phase cloud due to a drop of CCN may be a process that is a distinctive feature for mixed-phase clouds in the Arctic within a relatively pristine CCN environment. We do not know if this process is important for mixed-phase clouds in lower latitudes. However, clean layers with low CCN concentrations are also observed in marine stratocumulus clouds (e.g. [5, 1]) and generally this process can be important for all extremely-low CCN environments.

> Minor comment 1:
> P. 4, line 20, "smoothed from 12 km to 22 km": Are these
> numbers meaning altitudes?

Yes, these numbers are meaning altitudes. We have added the word height at the end of this sentences to clarify the meaning.

> Minor comment 2:
> Figures 3, 7, 9, and 10:  Did those profiles indicate model
> domain average?

These profiles indicate the horizontal mean at every vertical level shown at a certain time step. We have added "domain-averaged profiles" in every figure caption to avoid misunderstandings.

> Minor comment 3:
> P. 5, lines 21-22, "The $\theta$ profiles imply that ...":  I
> cannot see this in Fig.  4

That is a citing mistake. We apologize for the confusion. It is seen in Fig. 4 (previously Fig. 3) not in Fig. 5 (previously Fig. 4). We corrected it and cite Fig. 4.

> Minor comment 4:
> P. 6, line 7, "the LWP is slightly smaller (approximately 8
> g m$^{-2}$ after 4 h)":  I suppose that this sentence mentioned
> the CDNC 2 simulation.  If so, I think that a value of
> "8 g m$^{-2}$" in LWP is significantly smaller than that from
> the control simulation; it does not look like "slightly
> smaller".

We agree with the reviewer that the sentence was not clearly written. The "approximately 8 g m$^{-2}$ after 4 h" stands for the SensMoist simulation with dry air below the cloud (solid, pink line). This simulation is around 8 g m$^{-2}$ smaller than the 50 g m$^{-2}$ of the control simulation. Thus the LWP of the simulation with the dry air below the cloud is around 42 g m$^{-2}$. We changed the sentence to: "Up to 8 h, the LWP of the simulation with reduced moisture below the cloud is slightly smaller (by approximately 8 g m$^{-2}$ compared to the control simulation after 4 h) than in the control simulation and at the end of the simulation, the LWP is almost the same (Fig. 7, pink solid line)."

> Minor comment 5:
> I recommend referring a Heike Kalesse's (2016) paper
> titled "Understanding rapid changes in phase partitioning
> between cloud liquid and ice in stratiform mixed-phase
> clouds:  An Arctic Case Study".  This paper also mentioned a
> dissipation mechanism of Arctic mixed-phase cloud observed
> at Barrow, AK, on the basis of observation and partly model
> analysis. (Kalesse et al.  2016, Mon.  Wea.  Rev., doi:
> http://dx.doi.org/10.1175/MWR-D-16-0155.1)

Thank you for pointing out this paper to us. We have now considered it in the discussion section on page 8.

**Comments by Reviewer #2**

> General comment:
> ...  I would agree with Reviewer #3 that the lack of
> observational verification is a major limitation of this
> study.  Although the authors specifically state that their
> study exploits the observations collected during ASCOS,
> there is only a radar reflectivity plot and a moisture
> sounding presented in the paper.  I doubt detailed in-situ
> microphysical data are available for this case but perhaps
> some observations or reanalysis results characterizing the
> large-scale environment, which to a large extent determines
> the evolution of the cloud layer, could be included.

We have now included additional observational evidence in several locations throughout the manuscript and made relevant comparisons between these observations and model simulations. Please refer to the responses below and to the responses to Reviewer #3.

> Major comment 1:
> Since the CCN-limited regime is the main focus of the paper,
> I would expect that all simulations would be carried out
> using prognostic CCN instead of fixed CDNC. I consider this
> a major flaw that obviously warrants some discussion in the
> paper.  What are the potential implications and limitations
> of this approach?  For example, I would assume that with
> prognostic CCN the cloud layer would start diminishing at
> much higher background aerosol concentrations.

We have added the following sentences in section 3: "The advantages of this simplified approach, having fixed number concentrations of CDNC, is that the microphysical processes are constrained and can be easily varied in sensitivity experiments. The limitation is that the temporal evolution of a cloud layer consuming CCN by aerosol processing and scavenging can not be captured." Within the model intercomparison study in the BACCHUS project (www.bacchus-env.eu/), the COSMO model is performing simulations with prognostic CCN. Two different CCN concentrations are tested, $30\,\mathrm{cm}^{-3}$ and $80\,\mathrm{cm}^{-3}$. Both did not lead to dissipation of the Arctic mixed-phase cloud in the COSMO model simulations. This model study is done with several other models and is work in progress. It will be submitted later this year (Stevens et al., 2017, in prep.).

```
Major comment 2:
I'm not sure I understand the rationale behind the SensMoist
sensitivity simulations.  If the intent was to mimic a
large-scale drying advection, then why modify the initial
moisture profile and not just impose a forcing term
that would dry the domain out throughout the simulation?
Changing the initial moisture profile doesn't seem to
be a very realistic representation of drying large-scale
advection.  Is the magnitude of this drying consistent with
ECMWF reanalysis or following ASCOS soundings?
Also, how realistic is it to impose drying advection either
only below or only above the cloud layer?  Obviously, there
are two sources of moisture in the simulations - one above
the cloud layer and one below; therefore turning them off
one by one does not seem to be an effective way of turning
the moisture supply off.  Speaking of moisture sources, what
are the surface fluxes in the simulations?  I would also
strongly suggest including the vertical qv and moisture
turbulent flux profiles when discussing the simulations.
```

The intention of SensMoist was to study what happens to the Arctic mixed-phase cloud during the 31 August 2008 when drying the atmosphere either below or above the cloud and to study where (vertical) entrainment of air impacts the partitioning between vapor and cloud condensate. Thus, we mimic the change in the advection of air mass right at the beginning of the simulation. As in previous studies [10, 2], we are interested in the equilibrium state of the cloud and not in its temporal evolution. Hence, we think this setup is a fast way to mimic air mass changes. The SensMoist simulations generally indicate that the cloud condensate is more impacted by the lack of moisture above cloud top.

The surface fluxes are weak during the simulations, because we set the surface of the idealized simulations to a sea ice surface (Fig. 6). The sensible and latent heat flux are not able to strongly influence the boundary layer and the cloud development.

We have added following sentences in section 4.1:

"The sensible heat flux and the latent heat flux are weak, because the surface is covered with ice. The observed values of both of these fluxes are small, but positive [3]. Thus, no strong influence of the surface on the cloud is expected (Fig. 6)."

To show the differences in water vapor in the boundary layer we added a figure of domain mean profiles of QV at 5 h of simulation time and added the following sentences in section 4.2: "The mean profiles of the water vapor (QV) after 5 h of simulation show that the difference in moisture is small near the surface between the different simulations indicating a strong mixing in the sub-cloud layer (Fig. 8). A strong difference in QV is only seen above the cloud top and between the control simulation and the two SensMoist simulations with reduced RH."

[Figure]

Figure 2: Horizontal domain mean profiles of the moisture flux are shown after 5 h of simulation for the control simulation (red) and the SensMoist simulations (RH 10 %: yellow, dry below: green, RH 36 %:purple, and RH 20 %: blue).

The moisture flux at 5 h of simulation shows how large the differences are between the control simulation and the SensMoist simulations (Fig. 2, here in the reply text). The moisture flux is positive in the lower cloud layer in the control simulation as well as in the two SensMoist simulations with RH 36 % and RH 20 %. In the SensMoist simulation with dry air advection below the cloud, the moisture flux is generally smaller than all the other simulations and the vertical variation is also not as large as for instance in the control simulation, where a strong gradient of around -0.04 $\mathrm{m\,g\,s^{-1}\,kg^{-1}}$ is seen at around 500 m. However, the moisture flux fluctuates strongly with time. Thus, we have added only the QV profiles in the manuscript. These profiles show the effect of the moisture flux.

```
Major comment 3:
The analysis of SensCDNC simulations seems a bit dry.  I
would suggest expanding it, as this is the main point of
the paper.  Without knowing the simulations in detail, I
could speculate that the effect of decreasing CDNC is most
likely "indirect", i.e.  lower CDNC leads to lower LWC, and
consequently less radiative cooling, weaker cloud layer
circulation that cannot penetrate and mix the inversion at
300m, thus preserving the initial two-layer structure.  In
this configuration, the top cloud layer is isolated from
the moisture source below, the moisture source above is
weakened by the decreased entrainment and the cloud layer
begins to dissipate.  The bottom layer, however, seems quite
robust and does not show any signs of decay, which obviously
contradicts the observations in Fig.1.
```

Yes, you are right, decreasing the CDNC leads to an optically thinner cloud and
lower LWC, thus a weaker circulation in the liquid dominated cloud layer. The
description is found in section 4.4. Furthermore, we added some more sentences
in section 4.4.
"... due to the lower CDNC ..."
"The cloud-top radiative cooling is reduced, and subsequently the cloud-driven
circulation is unable to sufficiently penetrate the static stable layer near 300 m."

```
Minor comment 1:
Pg.  4 ln.5:  I assume that the horizontal grid spacing is
100m, not the model resolution.  Correct?  The same on ln.
7 for the vertical grid spacing.
```

Thank you for pointing that out to us. The term "resolution" is maybe not well
chosen. It is changed to grid spacing in the manuscript.

```
Minor comment 2:
What is the ice particle habit used in the simulations?  Is
it consistent with observations and/or temperature regime?
```

The velocity-mass and the diameter-mass relations are parameterized as described
in [4]. We assume dendrites in the model and the parameters in the velocity-mass
and the diameter-mass relations are adjusted to dendrites. The temperature in
the cloud layer ranges between $262\,\mathrm{K}$ and $265\,\mathrm{K}$. This is in the range of dendrites
and plates. Unfortunately, we do not have observations of the ice particle habit.

```
Minor comment 3:
Pg.  5 ln 30:  "When a cloud droplet grows ..." { perhaps
the model description section would be a better place for
this sentence?
```

Yes, it does not really fit at this point in the manuscript and the model setup is probably a better place. We moved it to section 3.

**References**

[1] D. Rosenfeld, Y. J. Kaufman, and I. Koren. Switching cloud cover and dynamical regimes from open to closed benard cells in response to the suppression of precipitation by aerosols. *Atmos.*, 6(9):2503–2511, 2006.

[2] J. Savre, A.M.L. Ekman, G. Svensson, and M. Tjernström. Large-eddy simulations of an arctic mixed-phase stratiform cloud observed during isdac : Sensitivity to moisture aloft, surface fluxes and large-scale forcing. *Q.J.R. Meteorol. Soc.*, July 2014.

[3] Joseph Sedlar, Michael Tjernström, Thorsten Mauritsen, MatthewD. Shupe, IanM. Brooks, P.OlaG. Persson, CathrynE. Birch, Caroline Leck, Anders Sirevaag, and Marcel Nicolaus. A transitioning arctic surface energy budget: the impacts of solar zenith angle, surface albedo and cloud radiative forcing. *Clim. Dyn.*, 37(7-8):1643–1660–, 2011.

[4] A. Seifert and K. D. Beheng. A two-moment cloud microphysics parameterization for mixed-phase clouds. part 1: Model description. *Meteorol. Atmos. Phys.*, 92(1-2):45–66, 2006.

[5] T. M. Sharon, B. A. Albrecht, H. H. Jonsson, P. Minnis, M. M. Khaiyer, T. M. van Reken, J. Seinfeld, and R. Flagan. Aerosol and cloud microphysical characteristics of rifts and gradients in maritime stratocumulus clouds. *J. Atmos. Sci.*, 63(3):983–997, 2006.

[6] M. D. Shupe, S. Y. Matrosov, and T. Uttal. Arctic mixed-phase cloud properties derived from surface-based sensors at sheba. *J. Atmos. Sci.*, 63(2):697–711, February 2006.

[7] M. D. Shupe, T. Uttal, and S. Y. Matrosov. Arctic cloud microphysics retrievals from surface-based remote sensors at sheba. *J. Appl. Meteor.*, 44(10):1544–1562, October 2005.

[8] Matthew D. Shupe. A ground-based multisensor cloud phase classifier. *Geophys. Res. Lett.*, 34(22), 2007.

[9] Matthew D. Shupe, Von P. Walden, Edwin Eloranta, Taneil Uttal, James R. Campbell, Sandra M. Starkweather, and Masataka Shiobara. Clouds at arctic atmospheric observatories. part i: Occurrence and macrophysical properties. *J. Appl. Meteor. Climatol.*, 50(3):626–644, 2011.

[10] A. Solomon, M. D. Shupe, O. Persson, H. Morrison, T. Yamaguchi, P. M. Caldwell, and G. de Boer. The sensitivity of springtime arctic mixed-phase stratocumulus clouds to surface-layer and cloud-top inversion-layer moisture sources. *J. Atmos. Sci.*, 71(2):574–595, October 2013.

[11] E. R. Westwater, Y. Han, M. D. Shupe, and S. Y. Matrosov. Analysis of integrated cloud liquid and precipitable water vapor retrievals from microwave radiometers during the surface heat budget of the arctic ocean project. *J. Geophys. Res. Atmos.*, 106(D23):32019–32030, 2001.